# Emissions of HFC-23 do not reflect commitments made under the Kigali Amendment
Ben Adam [1] ✉, Luke M. Western [1], Jens Mühle [2], Haklim Choi[3], Paul B. Krummel [4], Simon O'Doherty [1], Dickon Young [1], Kieran M. Stanley [1], Paul J. Fraser[4], Christina M. Harth[2], Peter K. Salameh[2], Ray F. Weiss [2], Ronald G. Prinn [5], Jooil Kim[2], Hyeri Park[6], Sunyoung Park[3,6] & Matt Rigby [1,5] ✉

HFC-23 (trifluoromethane) is a potent greenhouse gas released to the atmosphere primarily as a by-product of HCFC-22 (chlorodifluoromethane) synthesis. Since 2020, the Kigali Amendment to the Montreal Protocol has required Parties to destroy their HFC-23 emissions to the extent possible. Here, we present updated HFC-23 emissions estimated from atmospheric observations. Globally, emissions fell to $14.0 \pm 0.9$ Gg yr$^{-1}$ in 2023 from their maximum in 2019 of $17.3 \pm 0.8$ Gg yr$^{-1}$, but remained five times higher than reported in 2021. Atmospheric observation-based emissions for eastern China, the world's largest HCFC-22 producer, were also found to be substantially higher than 2020-2022 reported emissions. We estimate that potential HFC-23 sources not directly linked to HCFC-22 production explain only a minor, albeit highly uncertain, fraction of this discrepancy. Our findings suggest that HFC-23 emissions have not been destroyed to the extent reported by the Parties since the implementation of the Kigali Amendment.

The production and consumption of most hydrofluorocarbons (HFCs), which have replaced ozone-depleting substances in many applications, are controlled under the Kigali Amendment to the Montreal Protocol on Substances that Deplete the Ozone Layer[1]. This is due to the contribution of HFCs to global radiative forcing when emitted into the atmosphere[2]. While not yet universally adopted like the controls on ozone-depleting substances under the Montreal Protocol, the Kigali Amendment has currently been ratified by over 160 countries. The Amendment requires Parties to phase-down their HFC consumption; Article 5 (mostly developed) countries are required to reduce their consumption by 85% before 2036, while non-Article 5 (mostly developing) countries have until 2047 to reduce consumption by 80–85%. Additional controls are placed on HFC-23 (trifluoromethane, CHF$_3$), which Parties are required to destroy 'to the extent practicable' when it is formed as a by-product during the production of other HFCs or hydrochlorofluorocarbons (HCFCs)[1]. In the projects supported by the Multilateral Fund for the Implementation of the Montreal Protocol (MLF), destruction 'to the extent practicable' means that the mass of HFC-23 emitted should not exceed 0.1% of the mass of the target HFC or HCFC produced[3]. The requirement to abate HFC-23 emissions under the

Kigali Amendment began in January 2020, or the subsequent date when the Kigali Amendment was ratified by a particular Party. Here, we update (through to the end of 2023) earlier reported estimates of emissions of HFC-23[4–7], to investigate the impact of the controls imposed under the Kigali Amendment on emissions.

HFC-23 has the longest atmospheric lifetime (228 years) and the highest 100-year global warming potential (GWP$_{100}$ of 14,700) of the HFCs[8]. The Kigali Amendment permits the production of HFC-23 for a small number of uses[9], including as a feedstock for the production of halon-1301 (bromotrifluoromethane, CBrF$_3$) and for use in semiconductor etching, low-temperature refrigeration and fire suppression. Global consumption for such purposes is reported under the Kigali Amendment at ~1 Gg annually[9]. However, ~95% of the global production of HFC-23 is believed to be a by-product from the over-fluorination of HCFC-22 (CHClF$_2$) during its synthesis from chloroform (CHCl$_3$)[9]. HCFC-22 is a refrigerant and a feedstock used in the production of fluoropolymers and fluorochemicals[10]. Additional HFC-23 can be formed during the pyrolysis of HCFC-22 to tetrafluoroethene (TFE) and hexafluoropropene (HFP) during the production of polytetrafluoroethene (PTFE) and other fluoropolymers[11]. Other

[1]School of Chemistry, University of Bristol, Bristol, UK. [2]Scripps Institution of Oceanography, University of California San Diego, La Jolla, CA, USA. [3]Kyungpook Institute of Oceanography, Kyungpook National University, Daegu, Republic of Korea. [4]CSIRO Environment, Aspendale, VIC, Australia. [5]Center for Global Change Science, Massachusetts Institute of Technology, Cambridge, MA, USA. [6]Department of Oceanography, Kyungpook National University, Daegu, Republic of Korea. ✉e-mail: benjamin.adam@bristol.ac.uk; matt.rigby@bristol.ac.uk

sources of HFC-23 include its formation as a by-product during the industrial synthesis of various HFCs, including HFC-32 (difluoromethane, $CH_2F_2$), HFC-125 (pentafluoroethane, $C_2HF_5$) and others[9]. Further emissions may arise from incineration of HCFC-22-containing refrigeration equipment at end-of-life[12]. However, emissions from this source are yet to be quantified.

In addition to emissions from industrial activities, it has been suggested that HFC-23 may be generated in the atmosphere. Laboratory and modelling studies have indicated that HFC-23 is produced through the oxidation and subsequent photolysis of other HFCs, HCFCs and various hydrofluoroolefins (HFOs) via trifluoroacetaldehyde ($CF_3CHO$)[7,13,14]. Another study found that HFC-23 is produced in small quantities from the ozonolysis of some HFOs[15]. The contribution of the atmospheric generation of HFC-23 to the global HFC-23 burden is currently thought to be small (on the order of 0.5 Gg $yr^{-1}$)[7] but is projected to increase as HFOs are more widely adopted as replacements for HFCs[16].

Since the beginning of the first phase of the Kyoto Protocol in 2008, Annex I Parties to the United Nations Framework Convention on Climate Change (UNFCCC) have committed to reducing their emissions of a set of greenhouse gases that includes HFCs. Furthermore, these parties are required to report emissions of these gases to the UNFCCC annually. Reported emissions of HFC-23 from Annex I Parties have been small relative to top-down global emissions derived from atmospheric measurements in recent years[17], averaging ~1.5 Gg $yr^{-1}$ since 2009. Non-Annex I parties are not obliged to report their emissions of HFCs to the UNFCCC, but many have committed to reducing their HFC-23 emissions through other mechanisms. Between 2003 and 2014, under the UNFCCC Clean Development Mechanism (CDM)[18], payments were made from Annex I to non-Annex I Parties for the installation of HFC-23 abatement (destruction) equipment at HCFC-22 manufacturing facilities[19]. In addition, the two countries that produce the most HCFC-22, China and India, implemented national policies to reduce HFC-23 emissions. Since 2016, the Indian government has mandated that all HFC-23 by-product emissions be destroyed at HCFC-22 manufacturing facilities[20]. Similarly, under the HCFC production phase-out management plan (HPPMP) supported by the MLF, China reported abatement of 45, 93, 98 and 99.8% of HFC-23 emissions associated with HCFC-22 production in 2015, 2016, 2017 and 2018, respectively[21].

A previous study used bottom-up methods (i.e. a combination of HFC-23 emissions reported to the UNFCCC, HCFC-22 production reports and reported abatement) to estimate that global HFC-23 emissions were 2.4 Gg $yr^{-1}$ in 2017[4]. This reflected the ambitious targets of national abatement policies, such as China's HPPMP. However, top-down estimates based on atmospheric observations of HFC-23 mole fractions from the Advanced Global Atmospheric Gases Experiment (AGAGE)[22] suggested that, rather than declining to the levels estimated by the bottom-up approach, global emissions had instead increased to a historical high of $15.9 \pm 0.9$ Gg $yr^{-1}$ in 2018. These estimates were subsequently extended through 2022[5,7], and the discrepancy between the top-down and bottom-up emissions for that year was 10.5–12.5 Gg $yr^{-1}$.

Top-down estimates made using measurements from Gosan, South Korea[6] indicated that emissions of HFC-23 from eastern China in the period 2015–2019 represented a substantial fraction of the global total. Average emissions for eastern China of $7.2 \pm 0.4$ Gg $yr^{-1}$ were found during this period, with that region contributing $47 \pm 11\%$ of the global mismatch between top-down and bottom-up emissions estimates.

In this study, to determine any change in emissions since the implementation of the Kigali Amendment, we update HFC-23 mole fraction measurements and top-down emissions estimates globally[7] through 2023, and from eastern Asia[6] for the period 2020–2023. We also compare these to updated bottom-up emissions estimates[4,5,7] using emissions reported directly to UNEP by countries that have ratified the Kigali Amendment, and the most recent reports made to the UNFCCC. Finally, we explore by-product sources not directly linked to HCFC-22 production by considering emissions from the production of HFC-32, HFC-125 and TFE/HFP, as well

as in-atmosphere production via oxidation and ozonolysis. Through this investigation, we explore whether the Kigali Amendment-specified destruction of HFC-23 to the extent practicable during HFC and HCFC production has been achieved.

## Results

### Global emissions of HFC-23 through 2023

Background HFC-23 abundances measured in situ at five long-running AGAGE stations are assimilated into the AGAGE 12-box atmospheric model, and a Bayesian inversion method is applied to estimate global annual emissions through 2023 (see Methods section). Our estimates use a very similar methodology to previous work[4,5,7,23] and are an update using measurements through 2023. Our results indicate that HFC-23 emissions reached a maximum of $17.3 \pm 0.8$ Gg $yr^{-1}$ (1-sigma uncertainty) in 2019 and have since dropped by $3.2 \pm 1.3$ Gg $yr^{-1}$ ($19 \pm 8\%$), reaching $14.1 \pm 0.9$ Gg $yr^{-1}$ in 2023 (see Fig. 1a). These emissions drove a continued increase in the global mean mole fraction from 2020 to 2023, which was $36.8 \pm 0.9$ ppt in 2023 (see Fig. 1c). This mole fraction made a contribution to the direct radiative forcing (with stratospheric adjustment)[24] of $7.1 \pm 0.2$ mW $m^{-2}$, compared to a total from all HFCs of $42.3 \pm 1.3$ mW $m^{-2}$ for 2020[5].

### Emissions of HFC-23 from eastern Asia 2008–2023

To examine the drivers of the global trend over the period 2019–2023, we also recalculate emissions estimates from eastern Asia from 2008 through 2023 using observations of HFC-23 mole fractions measured at the AGAGE station in Gosan, South Korea. This is an update to previous work[6], using new measurements through to the end of 2023 and a revised inversion method (see Methods section). Measurements from this site allow estimation of emissions from eastern China, North Korea, South Korea and western Japan (as defined in the Methods section). We focus on this region because it is known to be important for HCFC-22 production and feedstock use[3], and because it is a region to which the atmospheric observation networks have sufficient sensitivity[25,26]. We find that emissions from this region reached a maximum in the period 2018–2019 and had decreased by ~30% in 2023. This fall was driven almost exclusively by a decrease in emissions from eastern China, where emissions dropped from $8.0 \pm 1.0$ to $5.6 \pm 0.7$ Gg $yr^{-1}$ between 2018 and 2023 (see Fig. 1a). This decrease is equivalent to $75 \pm 53\%$ of the global decline over the same period. Emissions from North Korea, South Korea and Japan were substantially smaller (averaging less than 0.6 Gg $yr^{-1}$ total across this period, as shown in Supplementary Fig. 1).

### Bottom-up estimates of HFC-23 emissions

Bottom-up estimates of HFC-23 emissions from HCFC-22 production are available from a variety of sources (see Methods). The sum of UNFCCC reports for Annex I countries and HCFC-22 production-based bottom-up estimates for non-Annex I parties suggest[7] that HFC-23 emissions globally remained at roughly 2–3 Gg $yr^{-1}$ for the period 2018–2022, after China's reported abatement under the HPPMP reached 99.8% (see Fig. 1a). These values are reasonably consistent with an independent bottom-up estimate of emissions from China of less than 1.0 Gg $yr^{-1}$ from 2018-2020[27]. The newly available emissions reported to UNEP combined with UNFCCC reports largely agree with the global HCFC-22 production-based estimates for the years when both are available. These reports totalled 2.8 Gg $yr^{-1}$ in 2021, the last year for which reporting is complete. The globally reported emissions in 2021 were dominated by China (1.1 Gg reported to UNEP) and Russia (1.1 Gg reported to the UNFCCC).

### Sources of HFC-23 not related to HCFC-22 production

The potential contribution to the global HFC-23 budget of sources not included in the UNFCCC and UNEP reports can be estimated based on emissions factors compiled by UNEP and other recently published data (see Supplementary Fig. 2). Non-HCFC-22 by-product sources include the generation of HFC-23 during HFC-125 production, HFC-32 production and TFE/HFP production. Furthermore, we consider HFC-23 formed in situ by oxidation and ozonolysis of other fluorinated compounds, such as

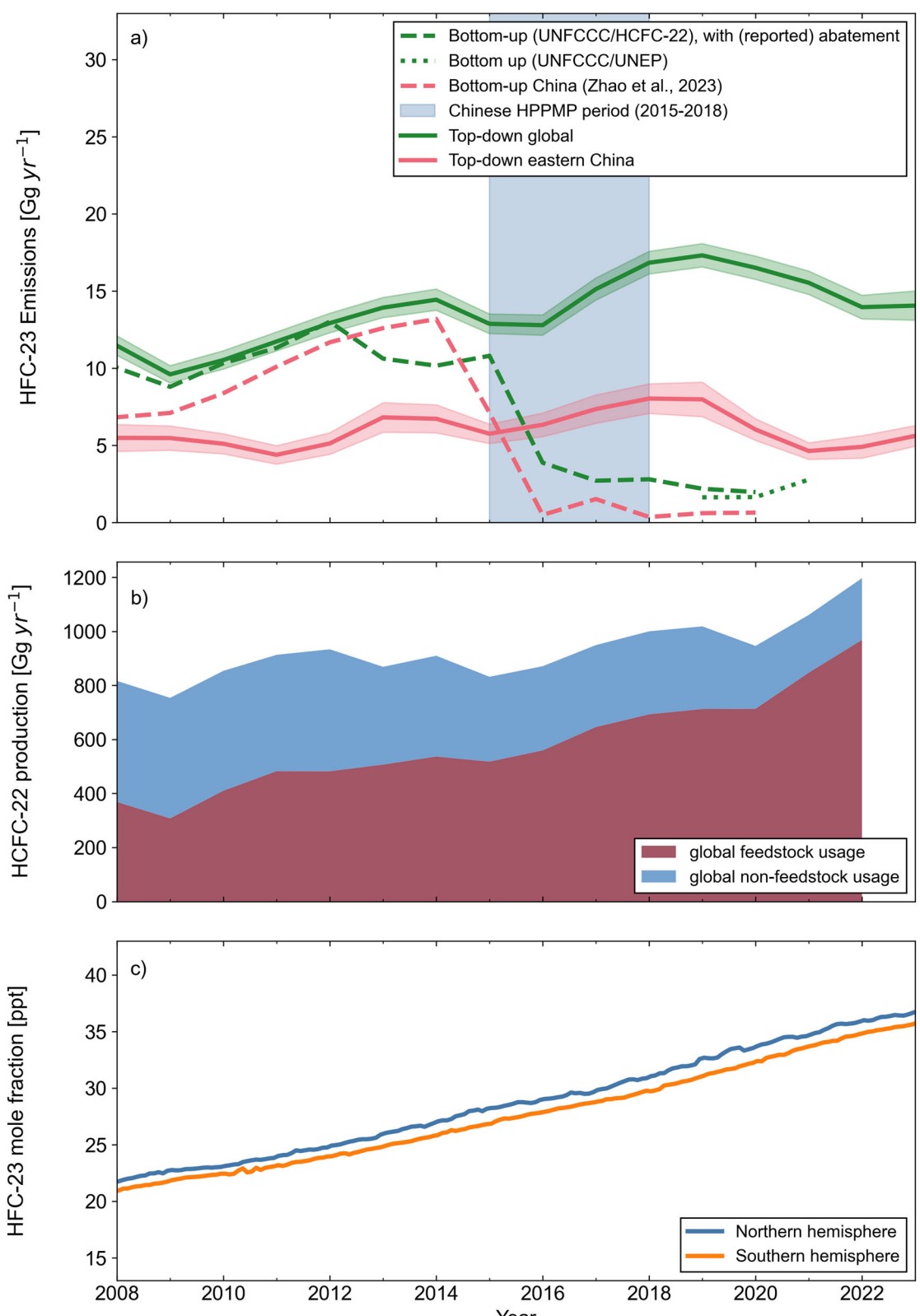

HFO-1234ze(E). For the industrial sources, we define low and high emissions scenarios, taking the smallest and largest values in the published ranges of emissions factors (see Methods). Of these sources, our estimates suggest that TFE/HFP production dominates (1.27 Gg yr⁻¹ in 2021, compared to 0.24 Gg yr⁻¹ for HFC-32 and 0.02 Gg yr⁻¹ for HFC-125 under the high emissions scenario). For HFC-23 production via oxidation and ozonolysis

in the atmosphere, we use the upper bound of 0.43 Gg yr⁻¹ from ref. 7. We find that even under this high emissions scenario, the other sources considered here lead to a bottom-up total (including emissions from HCFC-22 production and other reported sources) of 4.1 Gg yr⁻¹ in 2021, or 4.6 Gg yr⁻¹ including the oxidation and ozonolysis upper bound. Under the low emissions scenario, the total is 2.9 Gg yr⁻¹ for 2021. Even emissions from the

**Fig. 1 | HFC-23 emissions, HCFC-22 production and HFC-23 mole fractions, 2008–2023. a** Bottom-up and top-down HFC-23 emissions for the period 2008–2023. The green line shows the global emissions derived in this study from measurements of atmospheric mole fractions from the Advanced Global Atmospheric Gases Experiment (AGAGE). The pink line shows the emissions estimated from eastern China for this period, also derived in this study using data from Gosan, South Korea and the FLEXINVERT+ inversion framework. The green and pink shading represents the 1-sigma uncertainty in these estimates. The bottom-up estimates of global HFC-23 emissions through 2020 are shown by the dashed green line, based on emissions of HFC-23 reported to the United Nations Framework Convention on Climate Change (UNFCCC) and, for non-Annex 1 countries, HCFC-22 production data with abatement reported to the Multilateral Fund. These data suggest a substantial drop in emissions as China and India's abatement policies were implemented from 2015-2018. The green dotted line indicates emissions of HFC-23 from HCFC-22 production reported to the United Nations Environment Programme (UNEP) under the Kigali Amendment, combined with reports by Annex I countries to the UNFCCC for years where both are available; note that China only began reporting emissions to UNEP in 2020. The pink dashed line shows the bottom-up inventory of HFC-23 emissions for China taken from ref. 27. The blue shaded area indicates the period during which China's hydrofluorocarbon production phase-out management plan was in operation. **b** Global HCFC-22 production data for 2008-2022 for both feedstock (maroon) and non-feedstock (blue) uses, as reported to UNEP[53]. **c** Hemispheric monthly baseline mean HFC-23 mole fractions for 2008–2023, based on in situ measurements from the five core AGAGE sites, for the northern (blue) and southern (orange) hemispheres.

high emissions scenario are far lower than the top-down estimated emissions of $15.5 \pm 0.8$ Gg yr$^{-1}$ for 2021, and the discrepancy between the top-down and the updated bottom-up estimate is $10.9 \pm 0.8$ Gg yr$^{-1}$.

## Discussion

We find that the discrepancy identified in previous work[4,5,7] between the global bottom-up and top-down emissions has persisted in the four-year period after 2019, the first year of implementation of the Kigali Amendment. In 2021, the last year for which we have a complete set of emissions reports, this discrepancy was 12.7 Gg yr$^{-1}$. Bottom-up estimates remained roughly constant in the period 2018–2022, after China's reported abatement of HFC-23 emissions from HCFC-22 production reached 99.8%. Assuming no substantial change to these bottom-up estimates between 2022 and 2023 (which would be in line with the trend for 2018–2022), there remains a large gap between the top-down and bottom-up emissions estimates for 2023. Furthermore, top-down emissions from eastern China were $3.5 \pm 0.5$ Gg yr$^{-1}$ and $4.3 \pm 0.7$ Gg yr$^{-1}$ higher than were reported to UNEP for the whole of China in 2021 and 2022, respectively. This accounts for approximately one-third of the global discrepancy between the top-down and bottom-up estimates.

The bottom-up estimates and reported emissions assume that by-product emissions of HFC-23 from HCFC-22 production are nearly fully abated. Among the major HCFC-22 producers, China reported abatement at 99.8% from 2018 onwards, and India reported zero emissions from its HCFC-22 production facilities in 2021 and 2022, implying 100% abatement. Although total global reported HCFC-22 production for combined feedstock and non-feedstock uses has increased by approximately 25% since 2015[9,21] (see Fig. 1b), such abatement of HFC-23 emissions from this process would be expected to lead to a substantial drop in emissions of HFC-23 globally. This expectation assumes that HCFC-22 production is the main source of these emissions, which previous work showed to be the case before the implementation of abatement policies[4,23]. Our analysis of other potential sources of HFC-23 suggests that, based on currently available information, this is likely to still be the case. The magnitude of reported abatement is such that the increase in the amount of HFC-23 generated through HCFC-22 production should be outweighed by the near-total abatement, resulting in far smaller amounts released to the atmosphere overall than in 2015. The trend in global emissions derived from atmospheric data does not reflect this expectation, as emissions increased every year from 2016–2019 despite the increase in reported abatement, and remained above 2015 levels in 2023. This trend is also found in eastern China, where top-down emissions for 2021–2023 ($5.1 \pm 0.6$ Gg yr$^{-1}$ on average) are very similar to those in 2015 ($5.8 \pm 0.6$ Gg yr$^{-1}$), despite reported abatement increasing from 0% to 99.8% between these years. Therefore, if HCFC-22 production remains the dominant source of HFC-23 emissions in China, abatement levels must be lower than reported.

Based on the difference between our global and eastern Asian HFC-23 emissions estimates, we derive that $8.3 \pm 1.7$ Gg yr$^{-1}$ of HFC-23 emissions originated outside of eastern Asia in 2023. Only a small fraction of these can be attributed to a particular source location using top-down methods, due to the limitations of the spatial coverage of global monitoring networks[25]. Reported emissions from UNFCCC Annex I countries have remained low (<2 Gg yr$^{-1}$) since 2009, and where top-down estimates are available, there has not been evidence of dramatic under-reporting of emissions[28]. There are, however, several regions where emissions may have occurred in 2023, but to which the global measurement networks are currently insensitive. These include Russia, which reported HFC-23 emissions of similar magnitude to China in 2020 and 2021, and India, which reported zero emissions of HFC-23 in 2021 and 2022 despite operating multiple HCFC-22 production facilities. At present, top-down verification of these reports is not possible, and expansion of the measurement network to regions that are currently not observed would likely provide important new insights into the spatial distribution of HFC-23 emissions.

Our current understanding of emission factors suggests that sources of HFC-23 not directly linked to HCFC-22 production do not substantially reduce the discrepancy between top-down and reported global emissions. We find that emissions from HFC-32, HFC-125 and TFE/HFP production only explain a small fraction of the gap (0.4–1.5 Gg yr$^{-1}$), even in the case that emissions factors for these processes lie at the upper end of the published ranges. Therefore, we conclude that either emissions from HCFC-22 production are higher than reported, as discussed above, or that our understanding of the contribution of other sources is incomplete. It may be that emission factors from HFC or TFE/HFP production are far higher than stated, that atmospheric oxidation and ozonolysis play a much more important role than currently thought, or that other sources are missing in our analysis. Further work is required to better quantify these alternative sources.

Our results suggest a continued and substantial under-reporting of HFC-23 emissions from HCFC-22 production since the implementation of the Kigali Amendment. Whilst we report falling global emissions of HFC-23 during 2019–2023, during which period the Kigali Amendment was implemented by 160 countries, top-down emissions in 2023 were several times larger than those reported to the UNFCCC and UNEP. The fall in global emissions is largely driven by decreasing emissions from eastern China, but similarly to the global total, China's reported emissions are still many times lower than the regional top-down estimates based on atmospheric measurements. While the release of HFC-23 into the atmosphere during HCFC-22 production is likely to be the major contributor to global emissions, other sources could play a role. A bottom-up estimate presented here suggests that emissions of HFC-23 during the production of HFC-32, HFC-125 and TFE/HFP are likely to be small compared to those from HCFC-22 production, but a more complete quantification of these sources is needed. Strengthened reporting and monitoring of HFC-23 emissions from industrial activities is urgently required to better understand the discrepancy in the global HFC-23 budget, and its potential implications for the efficacy of the Kigali Amendment.

## Methods
### Bottom-up estimates and reports of HFC-23 emissions
Bottom-up estimates of HFC-23 emissions from HCFC-22 production for the period 2008-2022 were derived from several sources, as in previous work[4,5,7]. Emissions reported by Annex I Parties to the United Nations Framework Convention on Climate Change (UNFCCC) were taken from the 2023 National Inventory Reports (NIRs), covering the years 1990-2021[17]. Forty-three such reports were made in 2023, and reporting Parties

include the United States of America, Japan and most of Europe. These are disaggregated by sector and include emissions from the chemical industry, electronics industry and from use as substitutes for ozone-depleting substances in applications such as refrigeration, air conditioning and fire suppression. The total HFC-23 emissions reported under this framework is dominated by emissions from HCFC-22 production, which accounts for over 75% of emissions during each year of reporting. Reported HFC-23 emissions from sources not related to HCFC-22 production average less than 0.3 Gg yr$^{-1}$ globally throughout this period. In addition to these reports, Parties to the Kigali Amendment to the Montreal Protocol that produce HCFC-22 have been obligated to report their emissions of HFC-23 from HCFC-22 production directly to the United Nations Environment Programme (UNEP) since the ratification of the Kigali Amendment by that Party. Such emissions estimates are available for France, Germany, Japan, Mexico and the Netherlands from 2019–2023; North Korea from 2019–2022; Argentina from 2020–2023; China from 2020–2022; India and Russia from 2021–2023; Italy for 2022–2023; and South Korea for 2023[29]. In the case that countries report under both the UNEP and UNFCCC frameworks, there are only very minor (<0.001 Gg) discrepancies in almost all cases; the exception is for Russia, which reported emissions of ~1.1 Gg yr$^{-1}$ for 2019–2021 in its 2023 NIR but claimed zero emissions for 2021–2023 in reports to UNEP. Before reporting under the Kigali Amendment began (i.e. for 2008–2019 for China, for 2008-2020 for India), bottom-up emissions for non-Annex I Parties are estimated from HCFC-22 production data reported to UNEP (available through to 2021) and time-varying emissions factors (EFs). These EFs were derived from data reported by the MLF, which we extend through 2022 from previous work[4,5,7]. Here, we make a small revision to the estimates for 2013 and 2014 to rectify the double-counting of some abatement reports in this period.

## Inventory of other potential sources of HFC-23

In our extended bottom-up inventory, HFC-23 emissions estimates from HFC-32 and HFC-125 production were generated by combining HFC production estimates[30] with emission factor ranges reported to UNEP. The production was taken as the mean of the upper and lower bounds of the 'Kigali Amendment' scenario for 2021, and the emissions factors for our high and low scenarios were the upper and lower bounds (0.01-0.1% by weight for HFC-32, 0.001-0.01% by weight for HFC-125) published by the Technology and Economic Assessment Panel (TEAP) in 2023[9]. HFC-23 emissions during conversion of HCFC-22 to TFE/HFP are expected to be correlated with feedstock usage of HCFC-22, which is reported to UNEP and taken from ref. 10 for 2008–2019 and reports to UNEP for 2020 and 2021[31,32]. The high and low emissions scenarios took emissions factors from the Medical and Chemical Technical Options Committee (MCTOC) 2022 report, at 0.15 and 0.04% by weight, respectively[32]. Finally, we considered the contributions of the oxidation and ozonolysis of various fluorinated gases[7]. The most important source gases are HFO-1234ze(E) and HFO-1336mzz(Z), which produce HFC-23 through their reaction with OH and subsequent photolysis. The contribution from all source gases to the global HFC-23 burden was taken to be 0.43 Gg yr$^{-1}$, although this is an upper bound and the true contribution is likely to be lower. This is because the HFC-23 yields from photolysis experiments on which these calculations are based are presented as upper bounds, a result of the limit of detection of the instrument used. In addition, the HFO mole fractions used in the estimate were taken from measurements in Europe, a region in which HFOs have largely been phased in as HFC replacements. HFO mole fractions here are therefore higher than measured in remote sites, and unrepresentative of the atmosphere globally. The year 2021 was taken to be a representative year for this calculation, since it was based on atmospheric measurements of fluorinated compounds made between 2020 and 2023.

## Global measurements of HFC-23 mole fractions and emissions estimates

Global top-down emissions were derived for 2008–2023 using measurements made at five AGAGE stations: Mace Head (Ireland; 53.3°N, 9.9°W),

Trinidad Head (California, USA; 41.0°N, 124.1°W), Ragged Point (Barbados; 13.2°N, 59.4°W), Cape Matatula (American Samoa; 14.2°S, 170.6°W) and Kennaook/Cape Grim (Tasmania, Australia; 40.7°S, 144.7°E). These sites provide long-term background observations of a range of ozone-depleting substances and greenhouse gases, using Medusa gas chromatography-mass spectrometry (GC/MS) systems making high-frequency in situ measurements[22,33,34]. HFC-23 mole fractions are reported on the SIO-07 calibration scale, and measurements have been available since 2007 for this species. Earlier mole fractions are obtained from previous analysis of the Cape Grim Air Archive[33,35] and a small number of samples collected at Trinidad Head and other northern hemispheric sites. For further information, see ref. 4.

Monthly baseline averages of these measurements[36] were used to estimate global annual emissions, using the AGAGE 12-box atmospheric transport model[37,38] and a Bayesian inversion[39]. Briefly, the model divides the atmosphere along lines of latitude at 30°N, 0°N and 30°S to give zonal mean bands of equal mass, and a set of vertical divisions at pressures of 1000, 500 and 200 hPa. Transport between boxes is determined by seasonally averaged, annually repeating advection and diffusion parameters, and chemical loss in the troposphere is governed by reaction with a seasonally varying, but annually repeating, OH field[40] using a standard Arrhenius rate equation[41]. Year-to-year variations in large-scale atmospheric dynamics are not considered in this model, which may introduce errors in emissions estimates[42]. The stratospheric loss is modelled using a first-order rate constant, tuned to give a total global atmospheric lifetime of ~230 years, and uncertainty[43] in this value is accounted for in the final emissions estimates[44]. The inversion constrains the annual emissions growth a priori to 20% of the prior mean bottom-up emissions estimate. These prior mean estimates were taken from ref. 33 and repeated from 2010 onwards, consistent with previous global HFC-23 inversions[4,23]. A discussion of measurement uncertainties in both in situ and archive data can be found in ref. 4. The emissions estimates obtained differ slightly in places from previous work from which this study is an update[4,5], although the published estimates lie within ±1 standard deviation of the updated mean emissions for 10 of the 11 years 2008–2018. While this study uses the same historical measurements, slight differences in the reported mole fractions used as inputs in the model arise because of small adjustments to these measurements, e.g. due to updated calibration tank values and integration parameters. In addition, historical emissions are constrained and updated by new measurements from 2020–2023 to some extent under the Bayesian framework. Finally, a minor change was made to the inverse method between previous updates[4,5]. Nonetheless, the global emissions derived in this work show the same trends as in previous studies.

## Regional estimates of HFC-23 emissions from eastern Asia

The estimate of emissions in eastern Asia for 2008–2023 was performed using the FLEXINVERT+ Bayesian inversion framework[45], in an update to ref. 6. Small modifications were made to the inversion method from that study (as described below), and measurements from 2008–2023 were used to estimate emissions. FLEXINVERT+ is an analytical scheme that optimises posterior emissions on a variable grid. This grid has a resolution ranging from 0.5° × 0.5° to 12° × 12°, with finer resolution in regions where the sensitivity of measurements to emissions (as determined by the transport model) is highest. On this basis, we focus our analysis on an inversion domain bounded by lines of latitude at 20° and 50°N, and lines of longitude at 110° and 140°E. Within this domain, we focus on the regions of eastern China (defined here as the provinces of Anhui, Beijing, Hebei, Jiangsu, Liaoning, Shandong, Shanghai, Tianjin and Zhejiang), North Korea, South Korea and western Japan (defined as the prefectures of Chūgoku, Kansai, Kyūshū and Okinawa and Shikoku). In situ measurements of HFC-23 mole fractions were made using a Medusa GC/MS system at Gosan, South Korea (33.3°N, 126.2°E), and show little local influence due to the site's remote location at the southwestern tip of Jeju Island. The uncertainty associated with the measurements of HFC-23 mole fractions and the model is determined by the addition in quadrature of three terms; $\sigma_{inst}$, the instrumental precision based on the repeatability of the working standard, $\sigma_{bkg}$, the standard deviation of the monthly

 

background mole fraction used to determine the magnitude of the HFC-23 enhancement, and $\sigma_{model}$, a term used to account for model representation error, taken to be 8% of the monthly mean background mole fractions. The treatment of spatial and temporal error correlations is described in ref. [45]. The AGAGE statistical pollution filtering algorithm[36] is used to subtract the regional background mole fraction from each measurement and yield mole fraction enhancements above the baseline. These enhancements are shown in Supplementary Fig. 3. The enhancements are combined (without temporal averaging) with transport estimated from the FLEXPART[46] particle dispersion model, run for each measurement on a 0.5° × 0.5° grid and driven by hourly data from Climate Forecast System Reanalysis (CFSR) model meteorological reanalyses[47]. A priori emissions estimates were taken from ref. [48] (3.2 Gg yr$^{-1}$ for eastern China, 0.27 Gg yr$^{-1}$ for South Korea, 0.01 Gg yr$^{-1}$ for North Korea and 0.03 Gg yr$^{-1}$ for western Japan). The magnitudes of these estimates were kept constant throughout the inversion period. An ensemble of 36 model runs was defined using three a priori spatial distributions[6], three scaling factors applied to prior emissions and four flux error magnitudes applied to each variable resolution grid cell. These parameters are described in detail in the Supplementary Methods. For each inversion period, the final posterior emissions and uncertainties are calculated as the mean and standard deviation of the set of 36 mean posterior emissions estimates generated by the ensemble. The inversions estimated emissions over a period of two years rather than one, to reduce unrealistic interannual fluctuations[49]. In the emissions timeseries, the reported value for a given year is the mean of the emissions estimated from the two two-year inversions covering that year (i.e. the 2019 emissions are the mean of those from 2018 to 2019 and 2019 to 2020). Supplementary Fig. 4 shows the posterior spatial distribution of emissions for a selection of time periods. Typical error reduction in the posterior emissions was ~30–40%, and these are shown in Supplementary Fig. 5. As with the global estimates, these emissions estimates differ slightly from previously published emissions which have been updated here[6] due to updated historical measurements and the revised method. However, such deviations are small and the previously published estimates lie within ±1 standard deviation of the recalculated mean emissions for eight of the 12 years estimated (2008–2019), and within ±2 standard deviations for eleven of these.

## Reporting summary

Further information on research design is available in the Nature Portfolio Reporting Summary linked to this article.

## Data availability

Atmospheric measurement data from the AGAGE network are available at https://www-air.larc.nasa.gov/missions/agage/, last accessed 15 November 2024) and at https://doi.org/10.15485/2216951 ref. [50]. Derived global emissions estimates are in Supplementary Table 1, and eastern Asian estimates in Supplementary Table 2. Annex I National Inventory Reports to the United Nations Framework Convention on Climate Change are available from https://unfccc.int/ghg-inventories-annex-i-parties/2023 (last accessed 15 November 2024) and annual totals are in Supplementary Table 3. HFC-23 emissions reported under the Kigali Amendment are given in Supplementary Table 4 and are also available from https://ozone.unep.org/hfc-23-emissions (last accessed 15 November 2024).

## Code availability

The AGAGE 12-box model code (v0.2.2) is available via GitHub (https://github.com/mrghg/py12box, last accessed 15 November 2024 and https://github.com/mrghg/py12box_invert, last accessed 15 November 2024) and Zenodo (https://doi.org/10.5281/zenodo.6868589 ref. [51] and https://doi.org/10.5281/zenodo.6857794 ref. [52]). The FLEXINVERT+ code is available at https://git.nilu.no/flexpart/flexinvertplus.git (last accessed 15 November 2024).

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

## Acknowledgements

The authors particularly thank the continued cooperation and efforts of the station operators, who oversee the day-to-day running of the AGAGE sites. We thank the NASA Upper Atmosphere Research Program for its continuing support of AGAGE, including its five core stations, through grants 80NSSC21K1369 to MIT and 80NSSC21K1210 and 80NSSC21K1201 to SIO and earlier grants. Observations at Mace Head, Ireland, are partially supported by NASA and by the Department for Business, Energy & Industrial Strategy (BEIS, UK, formerly the Department for Energy and Climate Change), contracts 1028/06/2015 and 1537/06/2018 to the University of Bristol. Ragged Point, Barbados, is partially supported by NASA, and by the National Oceanic and Atmospheric Administration (NOAA, USA) through contract RA−133-R15-CN-0008 to the University of Bristol. Observations at Kennaook/Cape Grim, Australia, are supported largely by the Australian Bureau of Meteorology, CSIRO, the Australian Department of Climate Change, Energy, the Environment and Water (DCCEEW), NASA and

Refrigerant Reclaim Australia (RRA). B.A. is supported by the Natural Environment Research Council (NERC) GW4+ Doctoral Training Partnership. M.R. is supported by NERC grant NE/I021365/1. L.M.W. received funding from the European Union's Horizon 2020 research and innovation programme under the Marie Słodowska–Curie grant agreement no. 101030750. This research was supported by the National Research Foundation of Korea (NRF) grant funded by the Korean government (Ministry of Science and ICT; no. RS-2023-00229318). In addition, we are grateful for contributions from the Multilateral Fund Secretariat and the Ozone Secretariat.

## Author contributions

P.B.K., J.M., S.J.O., D.Y., K.M.S., P.J.F., C.M.H., J.K., P.K.S., R.G.P., R.F.W. and S.P. contributed observational data. B.A. and K.M.S. compiled the bottom-up inventory. B.A., M.R. and L.M.W. carried out global inverse modelling, and S.P., H.P., J.K. and H.C. performed the regional inverse modelling. B.A. wrote the paper, with contributions from L.M.W., J.M., M.R., H.C., and all co-authors.

## Competing interests

The authors declare no competing interests.
