## [Peer Review File · Communications Earth & Environment]

Emissions of HFC-23 do not reflect commitments made under the Kigali Amendment

Corresponding Author: Mr Ben Adam

This file contains all editorial decision letters in order by version, followed by all author rebuttals in order by version

This manuscript has been previously reviewed at another Nature Portfolio journal. This document only contains reviewer comments and rebuttal letters for versions considered at Communications Earth & Environment

Version 0:

Decision Letter:

Dear Mr Adam,

Your manuscript titled "Emissions of HFC-23 are higher than reported since the implementation of the Kigali Amendment" has now been seen by 2 reviewers, and we include their comments at the end of this message. They find your work of interest, but some important points are raised around clarity and discussions. We are interested in the possibility of publishing your study in Communications Earth & Environment, but would like to consider your responses to these concerns and assess a revised manuscript before we make a final decision on publication.

We therefore invite you to revise and resubmit your manuscript, along with a point-by-point response that takes into account the points raised. Please highlight all changes in the manuscript text file.

Please submit your point-by-point responses as a separate file, distinct from your cover letter where you can add responses to the Editors' comments that you do not want to be made available to the reviewers. Word files are preferred.

Important: The response to reviewers must not include any figures, tables or graphs. If you wish to respond to the reviewer reports with additional data in one of these formats, please add them to the main article or Supplementary Information, and refer to them in the rebuttal. Due to current technical limitations, any figures, tables, or graphs embedded in your rebuttal will not be included in the peer review file, if published.

Please use the following link to submit your revised manuscript, point-by-point response to the referees' comments (which should be in a separate document to any cover letter), a tracked-changes version of the manuscript (as a PDF file) and the completed checklist:

Link Redacted

We hope to receive your revised paper within six weeks; please let us know if you aren't able to submit it within this time so that we can discuss how best to proceed. If we don't hear from you, and the revision process takes significantly longer, we may close your file. In this event, we will still be happy to reconsider your paper at a later date, as long as nothing similar has been accepted for publication at Communications Earth & Environment or published elsewhere in the meantime.

Please do not hesitate to contact us if you have any questions or would like to discuss these revisions further. We look forward to seeing the revised manuscript and thank you for the opportunity to review your work.

Best regards,

Alice Drinkwater, PhD
Associate Editor

EDITORIAL POLICIES AND FORMATTING

Editorial Policy: [Policy requirements](https://www.nature.com/documents/nr-editorial-policy-checklist.pdf) (Download the link to your computer as a PDF.)

- Behavioural and social science
- Ecological, evolutionary & environmental sciences
- Life sciences

<https://www.nature.com/documents/nr-reporting-summary.zip>

Furthermore, please align your manuscript with our format requirements, which are summarized on the following checklist: [Communications Earth & Environment formatting checklist](https://www.nature.com/documents/commsj-phys-style-formatting-checklist-article.pdf)

and also in our style and formatting guide [Communications Earth & Environment formatting guide](https://www.nature.com/documents/commsj-phys-style-formatting-guide-accept.pdf) .

*** DATA: Communications Earth & Environment endorses the principles of the Enabling FAIR data project (<http://www.copdess.org/enabling-fair-data-project/>). We ask authors to make the data that support their conclusions available in permanent, publically accessible data repositories. (Please contact the editor if you are unable to make your data available).

All Communications Earth & Environment manuscripts must include a section titled "Data Availability" at the end of the Methods section or main text (if no Methods). More information on this policy, is available at <http://www.nature.com/authors/policies/data/data-availability-statements-data-citations.pdf>.

If a community resource is unavailable, data can be submitted to generalist repositories such as [figshare](https://figshare.com/) or [Dryad Digital Repository](http://datadryad.org/). Please provide a unique identifier for the data (for example a DOI or a permanent URL) in the data availability statement, if possible. If the repository does not provide identifiers, we encourage authors to supply the search terms that will return the data. For data that have been obtained from publically available sources, please provide a URL and the specific data product name in the data availability statement. Data with a DOI should be further cited in the methods reference section.

REVIEWER COMMENTS:

Reviewer #1 (Remarks to the Author):

The authors have clarified that they are updating the HFC-23 top-down emissions estimates since the ratification of Kigali, showing that the emissions gap reported by Park et al. 2023 has persisted past 2019. The findings of this top-down analysis will likely be useful to the Parties of the Protocol and the authors have addressed my concerns.

However, I don't believe this warrants an Article length publication in a Nature journal. The primary contributions of this paper are essentially 6 data points on one figure, (an update of previous analysis), and there are no meaningfully scientific contributions with respect to methods. I think a Brief Communication would be more appropriate.

Two small comments:

The bottom-up estimates stop in 2020. This is inconsistent with the title which emphasizes the discrepancy in reporting. I suggest revising the title to refer to Kigali commitments rather than reporting - especially since the authors have underscored that they are focused on "updating" previous work. The update is on top-down estimates, which was extended by 4 years, whereas the reporting was extended by 1 year.

Figure 2 could be moved to the supplement without detracting from the paper.

Reviewer #2 (Remarks to the Author):

The manuscript by Adam et al. presents an estimation of HFC-23 after the implementation of Kigali Amendment in Asian Countries. The estimation is very much required to understand the impact of Kigali Amendment and to provide policy directions to further Amendments. Determining the source of HFC-23 contribution is crucial and pretty challenging. The work presented is important in this aspect. However, on reading through the manuscript I feel that there is no proper flow. To understand the whole story, we need to go back and forth. I recommend major revisions addressing the following points before the paper is considered for publication.

1. The paper is lacking clarity in terms of discussions and also in explaining what exactly has been done. In particular, there is no clear distinction on what measurements have been performed, what are the data already obtained from secondary literature and what data exactly was used in the modelling studies.
2. The discussion that HCFC-22 production is the dominant source for HFC-23 is not convincing. The authors state that the production of HCFC-22 reduced by 25% between 2015 and 2021. Then again they state that HCFC-22 production is the dominant contributor for HFC-23. Both the statements are contradictory.
3. The authors claim that the formation of HFC-23 from other sources is very minor, without providing evidence. As the authors, mentioned in the introduction, the use of HFOs after 2015 may also be a source of HFC-23 through atmospheric ozonolysis. The authors can justify this by performing simple box model calculations and provide a kinetic estimate of formation of HFC-23 from HFOs. The estimation of HFC-23 emissions from ozonolysis of HFO-1234ze(E) is not clear and the corresponding scenarios developed need to be rechecked
4. Figure 2 is too confusing. The authors can only compare the top-down and bottom-up estimates here. Rest can be moved to SI

Minor corrections

1. Lot of typo errors – Example: Line 39
 2. The timeline of the phase-down of HFCs in China as per Kigali Amendment may be provided in the introduction section.
- Overall, the writing of the paper and the discussions need to be improved with clarity of the methods followed and the results obtained. The revised manuscript should be reviewed again.

Communications Earth & Environment is committed to improving transparency in authorship. As part of our efforts in this direction, we are now requesting that all authors identified as 'corresponding author' create and link their Open Researcher and Contributor Identifier (ORCID) with their account on the Manuscript Tracking System prior to acceptance. ORCID helps the scientific community achieve unambiguous attribution of all scholarly contributions. You can create and link your ORCID from the home page of the Manuscript Tracking System by clicking on 'Modify my Springer Nature account' and following the instructions in the link below. Please also inform all co-authors that they can add their ORCIDs to their accounts and that they must do so prior to acceptance.

Version 1:

Decision Letter:

Dear Mr Adam,

Your manuscript titled "Emissions of HFC-23 do not reflect commitments made under the Kigali Amendment" has now been seen by our reviewers, whose comments appear below. In light of their advice we are delighted to say that we are happy, in principle, to publish a suitably revised version in Communications Earth & Environment.

We therefore invite you to revise your paper one last time to edit your manuscript to comply with our format requirements and to maximise the accessibility and therefore the impact of your work.

EDITORIAL REQUESTS:

****Please take care to match our formatting and policy requirements. We will check revised manuscript and return manuscripts that do not comply. Such requests will lead to delays. ****

SUBMISSION INFORMATION:

OPEN ACCESS:

Communications Earth & Environment is a fully open access journal. Articles are made freely accessible on publication. For further information about article processing charges, open access funding, and advice and support from Nature Research, please visit <https://www.nature.com/commsenv/open-access>

Link Redacted

Best regards,

Alice Drinkwater, PhD
Associate Editor
Communications Earth & Environment
@CommsEarth

REVIEWERS' COMMENTS:

Reviewer #1 (Remarks to the Author):

The authors addressed my concerns.

Reviewer #2 (Remarks to the Author):

The authors have addressed all the comments raised earlier. The manuscript can now be accepted for publication in Communications Earth & Environment.

Kigali Amendment and to provide policy directions to further Amendments. Determining the source of HFC-23 contribution is crucial and pretty challenging. The work presented is important in this aspect. However, on reading through the manuscript I feel that there is no proper flow. To understand the whole story, we need to go back and forth. I recommend major revisions addressing the following points before the paper is considered for publication.

We thank the reviewer for their comments and have restructured the Discussion section to address this. The content in the first two paragraphs, which both pointed out the discrepancy between top-down and reported emissions, has been rewritten. These now read (L177-208 of the revised manuscript):

“We find that the discrepancy identified in previous work^{4,5,7} between the global bottom-up and top-down emissions has persisted in the four-year period after 2019, the first year of implementation of the Kigali Amendment. In 2021, the last year for which we have a complete set of emissions reports, this discrepancy was 12.7 Gg yr⁻¹. Bottom-up estimates remained roughly constant in the period 2018-2022, after China’s reported abatement of HFC-23 emissions from HCFC-22 production reached 99.8%. Assuming no significant change to these bottom-up estimates between 2021-2023 (which would be in line with the trend for 2018-2021), there remains a significant gap between the top-down and bottom-up emissions estimates for 2023. Furthermore, top-down emissions from eastern China were 3.5 ± 0.5 Gg yr⁻¹ and 4.3 ± 0.7 Gg yr⁻¹ higher than were reported to UNEP for the whole of China in 2021 and 2022, respectively. This accounts for approximately one-third of the global discrepancy between the top-down and bottom-up estimates.

The bottom-up estimates and reported emissions assume that by-product emissions of HFC-23 from HCFC-22 production are nearly fully abated. Among the major HCFC-22 producers, China reported abatement at 99.8% from 2018 onwards, and India reported zero emissions from its HCFC-22 production facilities in 2021 and 2022, implying 100% abatement. Total global reported HCFC-22 production for combined feedstock and non-feedstock uses has increased by approximately 25% since 2015^{9,21} (see figure 1b) and abatement of HFC-23 emissions from this process would be expected to lead to a significant drop in emissions of HFC-23 globally. This expectation assumes that HCFC-22 production is the main source of these emissions, which previous work showed to be the case before the implementation of abatement policies^{4,23}. Our analysis of other potential sources of HFC-23 suggests that, based on currently available information, this is likely to still be the case. The magnitude of reported abatement is such that the increase in the amount of HFC-23 generated through HCFC-22 production should be outweighed by the near-total abatement, resulting in far smaller amounts released to the atmosphere overall. The trend in global emissions derived from atmospheric data does not reflect this expectation, as emissions increased every year from 2016-2019 despite the increase in reported abatement, and remained above 2015 levels in 2023. This trend is also found in eastern China, where top-down emissions for 2021-2023 (5.1 ± 0.6 Gg yr⁻¹ on average) are very similar to those in 2015 (5.8 ± 0.6 Gg yr⁻¹), despite reported abatement increasing from 0% to 99.8% between these years. Therefore, if HCFC-22 production remains the dominant source of HFC-23 emissions in China, abatement levels must be lower than reported.”

We have also removed the next paragraph, which discussed the spatial distribution of emissions, as no meaningful conclusion was made and such results were not relevant to the overall argument. Finally, we removed reference to sources of HFC-23 not related to HCFC-22 production from the paragraph discussing alternative source regions for clarity. Other minor changes of wording and structure have been made throughout.

1. The paper is lacking clarity in terms of discussions and also in explaining what exactly has been done. In particular, there is no clear distinction on what measurements have been performed, what are

the data already obtained from secondary literature and what data exactly was used in the modelling studies.

In this study, we use measurements of HFC-23 mole fractions from the five core AGAGE sites for the period 2008-2023 to extend the box-model inversion to give global emissions for the period 2022-2023. In this regard, this work is an extension of Stanley et al. (2020), Liang & Rigby (2022) and Montzka et al. (2024). We have also updated measurements of HFC-23 mole fractions from Gosan, for the period 2019-2023 and used an updated inversion method to recalculate emissions for the period 2008-2023, extending and revising Park et al. (2023). The bottom-up inventory consists of emissions estimates based on HCFC-22 production data and emission factors previously reported in Liang & Rigby (2022) and Montzka et al. (2024). We have made further changes to the wording to clarify what data is included:

L111-115 of the revised manuscript reads “...we update HFC-23 mole fraction measurements and top-down emissions estimates globally⁷ through 2023, and from eastern Asia⁶ for the period 2020-2023. We also compare these to updated bottom-up emissions estimates^{4,5,7}, using emissions reported directly to UNEP by countries that have ratified the Kigali Amendment, and the most recent reports made to the UNFCCC.”

L121-125 of the revised manuscript reads “Background HFC-23 abundances measured *in situ* at five long-running AGAGE stations are assimilated into the AGAGE 12-box atmospheric model, and a Bayesian inversion method is applied to estimate global annual emissions through 2023 (see Methods section). Our estimates use a very similar methodology to previous work^{4,5,23} and are an update of Montzka et al.⁷ using measurements through 2023”

L132-136 reads “...we also recalculate emissions estimates from eastern Asia from 2008 through 2023 using observations of HFC-23 mole fractions measured at the AGAGE station in Gosan, South Korea. This is an update to previous work⁶, using new measurements through to the end of 2023 and a revised inversion method (see Methods section)”.

2. The discussion that HCFC-22 production is the dominant source for HFC-23 is not convincing. The authors state that the production of HCFC-22 reduced by 25% between 2015 and 2021. Then again they state that HCFC-22 production is the dominant contributor for HFC-23. Both the statements are contradictory.

The reviewer seems to have misunderstood this part of our study, so we have rephrased certain parts for clarity, as outlined below. For the avoidance of confusion: we find that emission of HFC-23, not HCFC-22 production, declined between 2015 and 2021. But in any case, this observed change on its own cannot tell us whether HCFC-22 production is the dominant source, since production and abatement will be changing simultaneously. Our finding is based on numerous considerations, as outlined in the paper: reported production, reported abatement, and current knowledge of potential unreported sources. Furthermore, we note that previous studies (e.g., Stanley et al., Miller et al.) found that before the end of the Clean Development Mechanism, top-down HFC-23 emissions agreed well with those estimated from HCFC-22 production alone, implying this was the dominant source historically. Our findings in this study suggest that, based on currently available information, other sources are unlikely to have changed this understanding.

L186-205 of the revised manuscript has now been changed to reflect this (see response to previous comment).

3. The authors claim that the formation of HFC-23 from other sources is very minor, without providing evidence. As the authors, mentioned in the introduction, the use of HFOs after 2015 may also be a source of HFC-23 through atmospheric ozonolysis. The authors can justify this by

performing simple box model calculations and provide a kinetic estimate of formation of HFC-23 from HFOs. The estimation of HFC-23 emissions from ozonolysis of HFO-1234ze(E) is not clear and the corresponding scenarios developed need to be rechecked

Since the original submission of this manuscript, another study (Montzka et al., 2024) has more thoroughly considered the atmospheric oxidation of a range of fluorinated gases to produce HFC-23, using a simple box model as the reviewer suggests. We have revised our bottom-up estimate to reflect this new information and acknowledge that this is just a single upper-limit.

L288-298 of the revised manuscript now reads “This was taken to be 0.43 Gg yr⁻¹, although this is an upper bound and the true contribution is likely to be lower. This is because the HFC-23 yields from photolysis experiments on which these calculations are based are presented as upper bounds, a result of the limit of detection of the instrument used. In addition, the HFO mole fractions used in Montzka et al. were taken from measurements in Europe, a region in which HFOs have largely been phased in as HFC replacements. HFO mole fractions here are therefore higher than measured in remote sites, and unrepresentative of the atmosphere globally. The most significant source gases are HFO-1234ze(E) and HFO-1336mzz(Z), which produce HFC-23 through their reaction with OH and subsequent photolysis. The year 2021 was taken to be a representative year for this calculation, since it was based on atmospheric measurements of fluorinated compounds made between 2020 and 2023..”

We also point out the need for future work in this area, and L229-230 of the revised manuscript now reads

“Further work is required to better quantify these alternative sources”

4. Figure 2 is too confusing. The authors can only compare the top-down and bottom-up estimates here. Rest can be moved to SI

We have re-formatted the figure to show only the contributions in 2021 as a bar chart, showing the ‘low’ and ‘high’ emissions scenarios. It has also been moved to the SI, as suggested. The updated figure and caption are presented below:

“Figure S1: Bottom-up inventory of HFC-23 emissions for 2021 under ‘low’ and ‘high’ emissions scenarios. Emissions from HFC-125 (dark blue) and HFC-32 (maroon) production are calculated using production data from Velders et al.¹ (see Methods) and the largest and smallest emissions factor from the range given by UNEP². Emissions from TFE/HFP production (light blue) are estimated using HCFC-22 feedstock usage from Mühle et al.³ and reports to UNEP^{4,5}, combined with the emissions factors range compiled by UNEP⁴. The emissions from HCFC-22 production (pink) are taken from

the reports to UNEP⁶ and the UNFCCC for 2021⁷, as with the data in Figure 1. The contribution from the atmospheric oxidation of fluorinated gases (yellow) to the ‘high’ emissions scenario is taken from the modelling work of Montzka et al.⁸, and contributes 0.43 Gg yr⁻¹, although this is reported as an upper bound and the true contribution is likely to be lower (see Methods). In the absence of a lower bound in that study, atmospheric oxidation and ozonolysis has been excluded from the ‘low’ emissions scenario. The estimate in Montzka et al. is based on measurements of fluorinated gases in the atmosphere for 2020-2023, so 2021 is chosen as a representative year.”

Minor corrections

1. Lot of typo errors – Example: Line 39

We thank the reviewer for pointing these out and have corrected them in the revised manuscript.

2. The timeline of the phase-down of HFCs in China as per Kigali Amendment may be provided in the introduction section.

L40-42 of the revised manuscript now reads “The Amendment requires Parties to phase-down their HFC consumption (developing countries by 80-85% by 2047, developed countries by 85% by 2036)”

Overall, the writing of the paper and the discussions need to be improved with clarity of the methods followed and the results obtained. The revised manuscript should be reviewed again.

We have endeavoured to improve the writing and clarity in a number of places, particularly in the Discussion. The paragraph discussing the spatial distribution of emissions has been removed. This is because its main finding, that the spatial distribution of HFC-23 emissions correlated well with the known locations of HCFC-22 factories, was not sufficient to discount the possibility that HFC-23 was being emitted from other industrial sources, which are largely co-located with HCFC-22 production facilities. Some discussion of this has been moved to the supplementary information. The discussion of other possible sources of HFC-23 emissions has been confined to one paragraph, to avoid confusion. Additionally, the first two paragraphs of the discussion have been restructured to provide a more concise summary of our findings.